# *Phaseolus acutifolius* Recombinant Lectin Exerts Differential Proapoptotic Activity on EGFR^+^ and EGFR^−^ Colon Cancer Cells and Provokes T Cell-Assisted Antitumor Responses in Mice

**DOI:** 10.3390/ph18020213

**Published:** 2025-02-05

**Authors:** Francisco Luján-Méndez, Patricia García-López, Laura C. Berumen, Guadalupe García-Alcocer, Roberto Ferriz-Martínez, Anette Ramírez-Carrera, Jaqueline González-Barrón, Teresa García-Gasca

**Affiliations:** 1Genetics and Biological Experimentation Laboratory, Faculty of Chemistry, Autonomous University of Querétaro, Querertaro 76010, Mexico; francisco.lujan@uaq.mx (F.L.-M.); lcbsq@yahoo.com (L.C.B.); 2Pharmacology Laboratory, Basic Research Subdirectorate, National Cancer Institute, Mexico City 14080, Mexico; pgarcia_lopez@yahoo.com.mx; 3Cellular and Molecular Biology Laboratory, Faculty of Natural Sciences, Autonomous University of Querétaro, Queretaro 76230, Mexico; roberto.augusto.ferriz@uaq.mx (R.F.-M.); anette.ramirez12@hotmail.com (A.R.-C.); jacky.gonbar09@gmail.com (J.G.-B.)

**Keywords:** colorectal cancer, colon cancer, recombinant lectin, *Phaseolus acutifolius*, biopharmaceutical candidate, T cells, apoptosis, immunogenic cell death, antitumor immune response

## Abstract

**Background:** *r*TBL-1, a recombinant lectin from *Phaseolus acutifolius*, exhibit proapoptotic activity on colon cancer cells and inhibitory properties on colon tumorigenesis in vivo. Apoptosis has been associated with a phospho-EGFR/phospho-p38/phospho-p53 mechanistic axis. Immunogenicity data have been observed in treated animals, but its possible involvement in the antitumor response remained unexplored. **Objective:** We investigated whether the cytotoxic activity of *r*TBL-1 depends on EGFR and its capacity to produce antitumor responses on syngeneic colon cancer in mice, with and without T cells, in order to explore its possible involvement in the process. **Results:**
*r*TBL-1 exhibited cytotoxic effects in a concentration-dependent manner in both EGFR^+^ (MC-38) and EGFR^−^ (CT-26) colon cancer cells with LC_50_ values of 23.50 and 30.01 µg/mL, respectively (*p* = 0.063). Apoptotic effects were slower and longer-lasting in MC-38 than in CT-26 cells. Significant increases in caspase-3 proteolytic activation and PARP1 cleavage were detected in both cell types, despite PARP1 rheostasis in CT-26 cells. Intralesional treatment with *r*TBL-1 inhibited the growth of established tumors in immunocompetent BALB/c mice in 27.81% (*p* = 0.0008) with a benefit in survival (*p* = 0.022), but not in immunodeficient BALB/c nude mice. **Conclusions:**
*r*TBL-1 induces apoptosis in colon cancer cells by EGFR independent mechanisms, although its presence could be related to deeper responses. Unresponsiveness in nude mice indicated that *r*TBL-1 antitumor effect is the synergistic result of apoptosis induction and T cell-mediated cytotoxicity in the tumor. Future studies will focus on the immunogenic effects triggered by the antitumor activity of *r*TBL-1 in colon cancer.

## 1. Introduction

Colorectal cancer (CRC) is one of the most common and lethal malignancies worldwide [1]. Although it is an etiologically heterogeneous condition, both in terms of its anatomical location and its molecular alterations, most cases of CRC are primarily linked to a series of modifiable risk factors typical of westernization, such as obesity, sedentary lifestyle, harmful dietary habits, smoking, and alcohol consumption, therefore, the increasing levels of which in developing countries could explain the growing burden of CRC [2].

As any oncological disease, CRC is product of the emergence of aberrant cell lineages that proliferate without being subject to external signals, forming ectopic, chronically proliferative, and eventually disseminative structures capable of establishing collaborative relationships with normal cells of the hematogenous and lymphatic vasculature, stromal cells, and cells of the immune system [3]. This ability contributes, under diverse molecular configurations, to the maintenance of CRC proliferative signaling by taking advantage of numerous microenvironmental sources of growth factors in a juxtacrine [4], autocrine, and paracrine manner [5,6]. Additionally, CRC proliferation is often driven by overexpression of tyrosine kinase receptors (TKRs) in their wild-type form or aberrant variants with hyperactivity due to oncogenic mutation. Such is the case of the epidermal growth factor receptor (EGFR), where its overexpression has been reported in a range of 25–82% of all cases of CRC [7,8], with a frequency of somatic mutations in around 3% of them [9]. Due to its relative suitability as a therapeutic target, EGFR inhibition remains a cornerstone in the treatment of metastatic CRC (mCRC) without downstream mutations from the target [10,11,12,13] and is are consolidated together with checkpoint inhibition immunotherapy indicated for the management of microsatellite instability-high (MSI-H) mCRC [14,15] as two of the main targeted therapies [16]. However, long-term survival in mCRC remains low [17], so efforts continue to develop new agents and combinations that enable more durable therapeutic responses and a better prognosis in these patients.

Our working group has developed *r*TBL-1 (MX/E/2023/021569), a recombinant lectin based on the coding sequence of a bioactive Tepary bean lectin (*Phaseolus acutifolius*) whose in vitro cytotoxic effects on a diversity of cancer cell lineages have been intensively studied [18,19,20], the same as its antitumor activity in CRC in vivo [21]. Recently, it has been validated that *r*TBL-1 retains the proapoptotic activity of its native counterpart on CRC cells in vitro (SW-480 and HT-29) in a manner linked to the recognition of tumor glycoconjugates through its carbohydrate binding pocket (CBP) [22], phosphorylation (Tyr1068), and lysosomal degradation of the EGFR, as well as phosphorylation of p38 (Thr180/Tyr182 [23]) and p53 (Ser15) [24].

In previous preclinical studies, we reported that intragastric administration of an enriched Tepary bean native lectins fraction (TBLF) whose majority component constitutes the lead compound of *r*TBL-1 [25], caused transient increases in the lymphocyte/granulocyte ratio, discrete increases in the splenic white pulp and at the level of the duodenum, and an increase in lymphatic follicles of Peyer’s patches, as well as in the concentrations of IL-6 in rats, data indicative of a secondary immune response to treatment [26,27,28]. However, the ability of *r*TBL-1 to produce antitumor responses in vivo has not been evaluated, and the possible involvement of the host immune system in such a response also remains unexplored.

In this work, we evaluated the ability of *r*TBL-1 to produce apoptosis in two murine colon cancer cell lines, with and without EGFR expression. Next, we evaluated the effect of the intralesional administration of *r*TBL-1 on the development of established EGFR^−^ ectopic tumor allografts in immunocompetent syngeneic rodents. Finally, using a reference model of athymic nude mice, we postulate the dependence on T cells of the *r*TBL-1-produced antitumor response.

## 2. Results

### 2.1. rTBL-1 Produces Cytotoxic Effects in a Concentration-Dependent Manner in Both EGFR^+^ and EGFR^−^ Murine CRC Cells

We previously reported that TBLF reduced the occurrence of low-grade premalignant lesions, concomitantly with a downregulation of the Akt pathway in colonic tissues of rats subjected to two chemically induced carcinogenesis protocols [21]. Subsequently, we demonstrated that *r*TBL-1 establishes supramolecular interactions matching at a significant level with EGFR on the surface of colon cancer cells, followed by lysosomal degradation of the receptor [24]. To determine whether the cytotoxic activity of *r*TBL-1 is dependent on EGFR in the cellular system, we evaluated the effect of a range of concentrations between 0 and 100 µg of *r*TBL-1/mL in two colon cancer cell types: MC-38 cells, EGFR^+^ [29], and CT-26, EGFR^−^ [30].

After carrying out its purification (Figure 1A,B) and once the identity of *r*TBL-1 in the sterile solution was confirmed by Western blot (Figure 1C), a concentration–response curve was obtained using serial dilutions with the appropriate assay medium (Figure 1D). At 8 h of exposure, MC-38 cells registered the greatest responsiveness to the treatment, presenting statistically significant drops at 20 µg of *r*TBL-1/mL (Dunnett, *p* = 0.0127). Meanwhile, at concentrations > 30 µg/mL, survival was reduced to less than 25% (*p* < 0.0001). For its part, although the response of CT-26 cells to treatment was slightly lower, their survival was significantly reduced at >30 µg/mL (*p* < 0.0001) to 54.03% ± 5.43, and greater concentrations produced the death of more than 85% of the initial cell population (*p* < 0.0001).

Comparison analyses between response patterns revealed that tolerance between cell lineages was significantly different at 30 µg of *r*TBL-1/mL (Student’s *t*-test, *p* = 0.011) (Figure 1E), showing a higher survival percentage in CT-26 cells than EGFR^+^ MC-38 cells. Finally, the dose–response curves (Figure 1F) confirmed that despite these differences in the response patterns, both tumor types responded to treatment with *r*TBL-1, although EGFR^−^ cells presented a 21.69% higher tolerance than EGFR^+^ cells in terms of their LC_50_, at 23.50 µg of *r*TBL-1/mL for MC-38 cells and 30.01 µg of *r*TBL-1/mL for CT-26 cells.

### 2.2. Proapoptotic Activity of rTBL-1 in Colon Cancer Cells MC-38 and CT-26

In previous studies, we have observed that the cytotoxicity induced by *r*TBL-1 in EGFR^+^ human colon cancer cells is related to apoptosis induction [24,31]. Since this is the first time that we have tested the bioactivity of *r*TLB-1 against murine colon cancer cells, as well as in an EGFR^−^ tumor type, we set out to evaluate the level of apoptosis induced by the treatment. We observed that at its LC_50_ values, *r*TBL-1 produces apoptotic differential response in MC-38 and CT-26 murine colon cancer cells after 8 h of exposure, without causing appreciable differences in cell necrosis (Figure 2A). The percentage of MC-38 cells in early apoptosis was 31.28% ± 3.23 (Student’s *t*-test, *p* < 0.005), 1.72 times more than control cells, similar to CT-26 cells, for which it was 1.6 times that with respect to control cells, although statistical significance was not reached, since the effects of treatment were directly reflected in late apoptosis, with 11.5 times more for EGFR^−^ CT-26 cells (*p* = 0.012) versus 5 times more for MC-38 cells (*p* = 0.001). Total apoptosis was increased 2.36- and 3.5-fold for MC-38 and CT-26 cells, respectively, compared to control cells. Control MC-38 cells showed a higher total apoptosis percentage of 22.76% ± 2.67 than CT-26 cells (13.26% ± 5.22) at 8 h.

Detailed analyses reveal that the apoptotic process induced by *r*TBL-1 in colon cancer cells is slower and longer-lasting in MC-38 EGFR^+^ cells than in CT-26 EGFR^−^ cells. After noticing a certain asynchrony in the progression of apoptosis induced by *r*TBL-1 in MC-38 and CT-26 tumor types, we set out to delve into their kinetics by comparing the cell subpopulations of both tumor types during early and late apoptosis, after 8 h of treatment. The analysis confirmed that, while the percentage of CT-26 cells in early apoptosis did not differ significantly from the vehicle-treated group (Figure 2B), MC-38 cells maintained a significantly higher representation with respect to control cells, as well as CT-26-treated cells (*p* = 0.005). These results confirm that, despite presenting a 21.69% lower tolerance to *r*TBL-1 (in terms of its LC_50_), MC-38 cells undergo apoptosis more gradually than CT-26 cells, whose response to treatment was notably more avid, consistently reaching the greatest cell subpopulation in late apoptosis in each assay. In addition, the observed low representation of CT-26 cells in early apoptosis suggests a cessation of the apoptotic response in the surviving subpopulation, unlike MC-38 cells, whose early apoptotic subpopulation persisted and was significantly elevated at 8 h of exposure to *r*TBL-1, suggesting continuity in the apoptotic process.

The comparison for late apoptosis between cell lines did not reach statistical significance (Figure 2C). Finally, no significant differences were observed between the percentages of MC-38 and CT-26 cells in total apoptosis post-treatment (Figure 2D). The difference observed between cell types exposed to the vehicle was statistically significant (*p* = 0.041) in early apoptosis, related to their sensitivity to deprivation of FBS.

### 2.3. rTBL-1 Induces Proteolytic Activation of Caspase-3 in Correlation with PARP1 Cleavage in Both Colon Cancer Types, Despite a PARP1-Rheostasis in the EGFR^−^ Phenotype

The nuclear protein Poly(ADP-ribose) polymerase 1 (PARP1) plays a key role in the activation of DNA repair complexes. High levels of PARP1 have been associated with increased proliferative competence in CRC cells, as well as decreases in overall survival (OS) and relapse-free survival (RFS) in patients with CRC [32]. PARP1 cleavage mediated by effector caspases 3 and 7 inhibits its activity, resulting in an accumulation of DNA lesions of lethal consequences for the cells [33]. Additionally, the 89-kDa fragment of PARP1 (cPARP1) amplifies the apoptotic stimulus through a variety of mechanisms at the cytosol level [34,35].

Treatment with *r*TBL-1 increased the cleavage of PARP1 in both MC-38 (Dunnett, *p* = 0.033) and CT-26 cells (*p* = 0.003), regardless of its EGFR status (Figure 3A,C). In correlation, significant increases in total caspase-3 levels were detected in both tumor types after *r*TBL-1 treatment (Figure 3D), with evident enrichments in the immunoblot of procaspase-3 (~31 kDa) and caspase-3 (17 kDa), suggesting increases in both expression and proteolytic activation of the effector caspase (MC-38, *p* = 0.004; CT-26, *p* = 0.042).

Interestingly, post-treatment levels of non-cleaved PARP1 were also significantly elevated in CT-26 cells (Dunnett, *p* = 0.038; Figure 3A,B), which might indicate an actionable expression of PARP1 or its stabilization by post-translational mechanisms [36]. Since CT-26 cells showed a slightly higher tolerance to *r*TBL-1, the elevation of PARP1 might represent a mechanism of resistance to treatment that correlates with the low representation of CT-26 cells in early apoptosis after 8 h of exposure to *r*TBL-1.

### 2.4. Intralesional Treatment with rTBL-1 Inhibits the Growth of Established Colon Cancer Tumors in Immunocompetent BALB/c Mice, but Not in Immunodeficient BALB/c Nude Mice

Having demonstrated that the proapoptotic activity of *r*TBL-1 is not restricted to EGFR-expressing cancers, and given that, unlike its native counterpart, *r*TBL-1 does not exhibit erythro-agglutinating properties [37] that contraindicate its administration by the intralesional route, we set out to investigate whether the proapoptotic activity of *r*TBL-1 is capable of producing antitumor responses in CRC in vivo in a targeted delivery setting. For this purpose, we chose CT-26 cells, whose genomic characterization accredits them as EGFR^−^, as well as being mismatch repair proficient, with a low neoantigen load and less immunogenic than MC-38 cells [30,38,39]. A total of 5 × 10^5^ viable cells were inoculated s.c. into haplotypically identical BALB/c (+/+) immunocompetent mice. On the other hand, to address the question of a possible immunological collaboration in the concretion of the antitumor response of *r*TBL-1, we conducted an analogous assay in BALB/c (*nu*/*nu*) hosts whose FOXN1^nu^ genotype determines the development of a dysfunctional rudimentary thymus, incapable of producing T cells [40].

After implantation, we evaluated the effect of an intralesional scheme of four doses of 6 mg of *r*TBL-1/kg in 50 µL of sterile PBS as vehicle on tumor growth inhibition (TGI) compared to that of a control group treated with the same volume of vehicle (Figure 4A).

BALB/c (+/+) mice presented a tumor growth kinetics that extended over three weeks, during which, with the sole exception of one individual in the control group, all animals completed the indicated administration schedule. One day after the third dose (day 12), the *r*TBL-1-treated group presented a significantly lower mean tumor volume than the control group (Student’s *t*-test, *p* = 0.005), followed by an asymptotic trajectory in the tumor growth curve (Figure 4B). On day 20, *r*TBL-1 treatment resulted in TGI of 27.81% (ratio paired *t*-test, *p* = 0.0008; confidence interval of the ratio from 0.6181 to 0.8310) with respect to the vehicle-treated group; no complete tumor regression was observed. Consistent with the TGI observed, treatment with *r*TBL-1 provided a survival benefit in BALB/c (+/+) mice (Log-rank [Mantel–Cox] test, *p* = 0.022; HR [logrank method] = 0.34), with 20% of the control group below the endpoint for tumor volume on day 24 (Figure 4D).

Tumor growth kinetics were twice as fast in BALB/c (*nu*/*nu*) mice, with 100% of animals reaching the tumor volume endpoint (>2000 mm^3^) in just 15 days. The benefit of *r*TBL-1 treatment was dependent on the availability of T cells in the host, so that no TGI was observed with respect to the group treated with vehicle (Figure 4C). Likewise, *r*TBL-1 administration did not provide benefit on the secondary outcome of survival of BALB/c (*nu*/*nu*) mice (Figure 4E), even despite the responsiveness previously observed in CT-26 cells treated in vitro.

## 3. Discussion

The cytotoxic properties of TBLF, obtained from the seeds of *Phaseolus acutifolius*, have been the subject of numerous studies carried out by our group and others. Accumulative evidence supports the inhibitory effects of TBLF on nine different cancer cell lines (MCF7, derived from breast adenocarcinoma; ZR-75-1, from breast ductal carcinoma; HeLa, from cervical adenocarcinoma; C-33 A and SiHa, from cervical carcinomas; RKO, from colon carcinoma; and derived from colorectal adenocarcinomas, Caco-2, HT-29, and SW-480 [18,19,20]), whereas the tolerance observed in cells without oncogenic transformation was ≈10 times higher [19]. At the in vivo level, intragastric administration of TBLF (50 mg/kg every 72 h, for six weeks) revealed a good tolerability profile in rats [27], as well as the ability to produce an inhibitory effect on early colon carcinogenesis through two chemical induction schemes, observing a 50% reduction in the occurrence of aberrant crypt foci in animals subjected to a protocol with azoxymethane + dextran sodium sulfate (AOM/DSS), in concomitance with a downregulation of Akt phosphorylation (Ser473) in the treated tissues and an appreciable increase in lymphocytic infiltration at the colon level [21].

Following the establishment of a system for the heterologous production of the major lectin in TBLF (TBL-1) and the resulting production of *r*TBL-1 [25], Dena-Beltrán et al. [24] observed that exposure to concentrations starting from 61 µg/mL for 24 h was effective in replicating the apoptotic effects of TBLF on SW-480 cells, derived from a locally advanced Dukes’ type B colorectal adenocarcinoma [41] with high expression of wild-type EGFR [42], in terms of proteolytic activation of caspase-3, cleavage of PARP-1, and phosphorylation (Ser139) of histone HA2X [43].

Further analysis by immunofluorescence revealed a colocalization of *r*TBL-1 and EGFR on the cell surface, followed by an increase in phosphorylation (Tyr1068) and lysosomal degradation of the receptor. Downstream, increases in phosphorylation of p38 kinase (Thr180/Tyr182) as well as in the transcriptional regulators STAT1 (Tyr701) and p53 (Ser15) were detected. Causally associated with the cell cycle arrest and apoptosis observed in treated SW-480 cells, the increases in phospho-STAT1 and phospho-p53 led to the postulation of a possible biased signaling of *r*TBL-1 via EGFR [24]. In this same sense, CHO-K1 cells (derived from Chinese hamster ovary) negative for EGFR expression [44] were markedly refractory to the effects of *r*TBL-1, with no proteolytic activation of caspase-3 detected in cells exposed to concentrations of up to 122 µg/mL for 24 h. However, given the expected phenotypic differences that a non-tumor cell type implies, the validity of CHO-K1 cells as an EGFR^−^ reference model could be arguable. Initially, in this work, we proposed to evaluate whether the proapoptotic activity of *r*TBL-1 on CRC is dependent on the availability of EGFR in the cellular system.

Our results indicate that colon cancer cells, both EGFR^+^ (MC-38) and EGFR^−^ (CT-26), are responsive to 8 h in vitro treatment with *r*TBL-1, presenting LC_50_ values of 23.50 and 30.01 µg/mL, respectively (*p* = 0.063). Although CT-26 EGFR^−^ cells showed a tolerance 21.69% higher than that observed in MC-38 EGFR^+^ cells, this represents only 53.59% of the previously reported *r*TBL-1-LC_50_ of HT-29 cells (56 µg/mL) [31], whose wild-type EGFR expression levels [45] are approximately equal to those observed in normal cells [46].

It is noteworthy that, within the four colon cancer cell types that have so far been confirmed in vitro as *r*TBL-1-responsive, only MC-38 cells harbor a mutation in EGFR (G1103C) at the level of the C-terminal tail [29]. However, it is thought to be a passenger mutation since, as far as is known, it has no oncogenic character [47]. Additionally, since this mutation results in an alteration in the strictly intracytoplasmic C-terminal end, its involvement in the molecular interactions established with *r*TBL-1 in the extracellular environment is unlikely.

In addition, it is known that the oncogenic alteration of a variety of EGFR downstream effectors releases them from its regulation and determines both their unbridled signaling as well as a cellular proliferation without subjection to the receptor and its ligand [7]. In this regard, oncogenic mutations in GTPase proteins of the RAS family, as well as in the serine/threonine kinase BRAF, stand out as underlying conditions for proliferative autonomy in 52 and 10% of all CRC cases, respectively [48,49]. As such, clinical detection of RAS and BRAF mutational statuses constitute prominent markers to guide therapeutic choice in mCRC [50], while in cases with wild-type RAS/BRAF expression, EGFR constitutes a therapeutic target par excellence [12]. In this regard, considering the oncogenic mutational statuses of KRAS and BRAF harbored in colon cancer cell lines with confirmed responsiveness to *r*TBL-1 (Table 1), a therapeutic approach based on blocking or inhibiting EGFR would be an ineffective strategy, since in such models, the signaling is transduced even in the absence of any upstream signals. Therefore, the EGFR-independent mechanistic components in the proapoptotic activity of *r*TBL-1 remain to be identified.

On the other hand, in accordance with the effects previously reported in other cancer cell types, the inhibitory properties of *r*TBL-1 on MC-38 and CT-26 cells correspond to a potent apoptotic effect, with a 2.36- and 3.5-fold increase in total apoptosis, respectively, after 8 h of exposure to the corresponding LC_50_-*r*TBL-1.

The differences observed between the apoptotic effects of each tumor type demonstrate that MC-38 cells exhibited slower and lasting apoptosis than that observed in CT-26 cells, whose level of apoptosis was detected mainly in the late phase but ceased after 8 h of exposure, as demonstrated by the low representation of cells in early apoptosis. In this regard, previous studies indicate that *r*TBL-1 binds specifically to β1-6 branched *N*-glycans’ associated with cancer [25], and more recently, Martínez-Alarcón [22]. extended their observations by predicting, through molecular docking analysis, high association affinities between *r*TBL-1 and various aberrantly expressed *N*-glycans in the EGFR of tumor cells in a dependent manner on *r*TBL-1 CBP residues [22]. In this sense, it might be reasonable to hypothesize that, given their EGFR^+^ phenotype, MC-38 cells experienced a higher level of *r*TBL-1 binding than CT-26 cells, possibly determining the cross-linking of EGFR monomers to other surface receptors (also equipped with carbohydrate motifs cognate to the CBP of *r*TBL-1), whose collective interaction resulted in a pro-apoptotic stimulus of longer duration than in CT-26 cells, whose surface glycan availability would not include those of the EGFR.

In accordance, Parshenkov and Hennet [59] studied the effects of a variety of plant and fungal lectins on MC-38 cells, finding that wheat germ agglutinin, *Maackia amurensis* lectin, I and *Aleuria aurantia* lectin induced various cell death pathways through cross-linking of the glycosylated receptors Fas-associated death domain (FADD), tumor necrosis factor receptor type 1-associated death domain protein (TRADD), and Toll-Like Receptors (TLRs), leading the treated cells to resolutions by extrinsic apoptosis, necroptosis, and pyroptosis, respectively [59]. In an analogous but opposite manner, lectins such as concanavalin A can support cross-linking of glycosylated TKRs with mitogenic outcomes in T cells [60].

Under this framework, it is possible that the cross-linking of EGFR monomers produced by *r*TBL-1 treatment at the cell surface level could have enabled its intermolecular autophosphorylation [61] and possibly induced the activation in parallel of some form of receptor-mediated cell death. However, these observations require adequate experimental demonstrations.

On the other hand, the analysis of the apoptosis markers produced after *r*TBL-1 treatment confirmed significant increases in both the proteolytic activation of caspase-3 and the inactivation by cleavage of PARP1 in both cell types. Interestingly, although basal levels of full-length PARP1 were equivalent in control MC-38 and CT-26 cells, a significant elevation was detected in the latter in response to *r*TBL-1 treatment. Since actionable increases in PARP1 in CRC are related to the IL-6/Phospho (Tyr705)-Stat3 axis [62], which promotes cell proliferation and survival mediated by cyclin D1 and BCL-xL, respectively [63], such a response could constitute a rheostatic maneuver of CT-26 cells to treatment, possibly related to their greater tolerance, as well as to the shorter duration of the apoptotic stimulus observed in this cell type. Together, these results indicate that although the ability of *r*TBL-1 to induce apoptosis in CRC cells does not strictly depend on the abundance or availability of EGFR in the cellular system, its expression could be associated with deeper responses. Therefore, preclinical evaluation of *r*TBL-1 in CRC is relevant, regardless of the level of EGFR expression in the tumor.

However, it should be noted that HT-29 (EGFR^+^) cells treated with the corresponding TBLF or *r*TBL-1 LC_50_ showed similar increases in their full-length PARP1 levels [24], which might suggest that the EGFR expression in the cellular system is not the direct or unique underlying cause. Currently, the use of PARP1 inhibitory agents as sensitizers together with systemic chemotherapy is contemplated for the management of CRC with homologous recombination repair (HR) deficiency [64]. Therefore, a joint use of *r*TBL-1 and PARP1 inhibitors could produce a synergistic response in these models.

On the other hand, the data available on the therapeutic activity of *Phaseolus acutifolius* lectins in animal models of CRC until now were limited to those of TBLF, obtained and informed by our group. Here, we report the first evidence of antitumor activity of *r*TBL-1 on CRC in an allotransplantable mouse model. We also demonstrate that the contribution of T cell-mediated immunological mechanisms is a crucial aspect for the concretion of the in vivo antitumor response of *r*TBL-1.

Previously, a cluster of preclinical evidence demonstrated that TBLF exhibits, as well as multiple lectins produced in plants of the *Fabaceae* family [65], erythroagglutinating properties on glutaraldehyde-fixed rabbit or type A+ human erythrocytes, as well as considerable resistance to digestion [26,27]. These factors were decisive for the evaluation of the therapeutic potential of TBLF in vivo to be developed through intragastric administration and on colon cancer, whose anatomical location conferred relative suitability for its experimental approach with TBLF. For its part, *r*TBL-1 partially resists digestive conditions but does not have erythroagglutinating activity [37]; therefore, preclinical exploration of its effects could potentially be extended to other solid tumors, not limited to the digestive tract. However, more data are needed regarding the safety profile of *r*TBL-1 administered by alternative routes.

In this work, we tested the efficacy of a four-dose intralesional scheme of *r*TBL-1 (6 mg/kg) to produce an antitumor response in subdermal ectopic masses of syngeneic CT-26 colon cancer, established in BALB/c (+/+) mice. The treatment produced a TGI of 27.81% (*p* = 0.0008) with a modest but significant benefit in 27-day survival (*p* = 0.022). Since CT-26 cells are derived from a grade-IV undifferentiated colon carcinoma, rapidly growing and highly metastatic [66], these results indicate that intralesional administration of *r*TBL-1 could contribute to improving efficacy in more advanced stages of the disease.

On the other hand, immunodeficient BALB/c (*nu*/*nu*) mice subjected to the same treatment did not experience TGI or improvements in survival 15 days after the start of the assay. Such disparity in the treatment responses seems to indicate that the antitumor effect of *r*TBL-1 constitutes the synergistic result of the induction of apoptosis in the tumor mass and T cell-mediated cytotoxicity. In this regard, professional antigen-presenting cells (APCs) of various lineages [67,68,69] can acquire the ability to capture tumor antigens, not only in the context of inflammation inherent to certain cancers [70] but also as a result of certain therapeutic interventions [71,72,73]. In such cases, the derived peptide antigens are presented to T cell receptors, expressed on CD4^+^ [74] and CD8^+^ cells [75], together with several accessory coreceptors that collectively sustain their activation as CD4^+^ helper T cells type 1 (T_h_1) and cytotoxic T lymphocytes (CTLs), endowed with the capacity to amplify and execute antigen-specific antitumor immune responses, respectively [76]. In this sense, post-treatment analyses related to the possible modulation of infiltrating immune subpopulations could not be performed in this work because the tumor volumes reached caused a generalized presence of necrosis in the specimens, severely compromising their analytical usefulness. Another limitation experienced was the loss of up to two individuals from the same experimental group due to the establishment of two simultaneous tumors in the same host and the impossibility of the experimental approach by means of intralesional injection.

In light of our results, it should be noted that the intragastric administration of TBLF in healthy Sprague Dawley rats led to increases in the serum concentration of IL-6 in the splenic white pulp, as well as in the lymphatic follicles of the Peyer’s patches in the duodenum [27]. While these effects could be the result of TBL-1 and other proteins contained in TBLF operating as immunogens, the increase in lymphocyte infiltration observed in the colon of rats challenged with AOM/DSS and treated with TBLF could have been indicative of the induction of a T_h_1 response, given the observed apoptotic activity in precancerous lesions [21]. Future studies will focus on elucidating the immunogenic effects triggered by the antitumor activity of *r*TBL-1 in CRC.

## 4. Materials and Methods

### 4.1. rTBL-1 Production and Purification

This material was obtained from the liquid culture of a *Pichia pastoris* strain previously transformed for the heterologous expression of *r*TBL-1 [25] and purified following an adaptation [31]. In a first step, *r*TBL-1 was recovered from the previously concentrated culture medium by tangential flow (SartoJet Pump, Sartorius Stedim Biotech GmbH Cat. Num. 17521-110, Goettingen, Germany) and subsequently purified by column chromatography based on nickel bioaffinity (11197. Ni Sepharose 6 Fast Flow 45-165UM, GE HEALTHCARE Cat. Num. 17531802, Darmstadt, Germany) and semipermeable membrane dialysis at 4 °C. Finally, the *r*TBL-1 was lyophilized (0.125 Scientz-10N, Ningbo Scientz Biotechnology Co., LTD., Ningbo, China) and stored at temperatures of 0 to 4 °C in 50 mL conical tubes. Prior to its use, 50 µg of anhydrous *r*TBL-1 was reconstituted in 5 mL of PBS and the resulting solution was sterilized through 0.22 µm membrane filtration (NEST Scientific Cat. Num. 380111, Wuxi, China). The protein concentration was determined by nanodrop (Thermo Scientific™ Cat. Num. ND-ONE-W, Wilmington, DE, USA) and the identity of the molecule in each batch was confirmed by denaturing electrophoretic analysis in 10% sodium dodecyl sulfate polyacrylamide gel (SDS-PAGE) [77] and stained with Coomassie blue (Thermo Fisher Scientific™ Cat. Num. 20278, Rockford, IL, USA), followed by Western blotting for the 6xHis tag of *r*TBL-1 (6x-His Tag Monoclonal Antibody [HIS.H8] Invitrogen^®^ Cat. Num. MA1-21315, Vilnius, Lithuania),in a concentration of 1:1000, and as secondary antibody, peroxidase-conjugated anti mouse IgG (H+L) in donkey (Jackson Immunoresearch Lab. Inc. Cat. Num. 715-035-150, West Grove, PA, USA) in a concentration of 1:10,000 which, each time, showed a single band of molecular weight of ~30.8 kDa.

### 4.2. Cell Lines

Colon cancer cells: the CT-26 line (ATCC^®^ Cat. Num. CRL-2638™ Manassas, VA, USA) derived from undifferentiated murine colon carcinoma induced by N-nitroso-N-methylurethane (BALB/c strain, H2^d^), which constitutes a non-microsatellite instable model [39] negative for the expression of EGFR [30,57], and the MC-38 line (Kerafast, Inc. Cat. Num. ENH204-FP, Boston, MA, USA), derived from murine colon adenocarcinoma (C57BL6 strain, H2^b^) induced by methylcholanthrene, which, in addition to being positive for EGFR expression [29,57], has been validated as a colon cancer with deficient mismatch repair (dMMR) and MSI-H [39], were used in this work. Cells were stored into cryovials in the vapor phase (−190 °C) of a liquid nitrogen tank until cultivation and propagation in the Genetic Research and Biological Experimentation laboratory at the School of Chemistry of the Autonomous University of Querétaro. In accordance with the specifications of each line, CT-26 cells were cultured in a growth medium of ATCC modified RPMI-1640 (GIBCO^®^ Cat. Num. A1049101, Miami, FL, USA) supplemented with 10% fetal bovine serum (FBS; GIBCO^®^ Cat. Num. 26140079) and 1% Anti-anti (GIBCO^®^ Cat. Num. 15240-062). MC-38 cells were grown in DMEM with high glucose and pyruvate content (GIBCO^®^ Cat. Num. 11995065) supplemented with 10% FBS, 0.1 mM non-essential amino acids (Corning^®^ Cat. Num. 25 -025-CI, Durham, NC, USA), 10 mM HEPES buffer (Corning^®^ Cat. Num. 25-060-CI), and 1% Anti-Anti (GIBCO^®^ Cat. Num. 15240-062, Miami, FL, USA). Finally, the cultures were placed in incubation (Thermo Scientific™ Model 3110, Boston, MA, USA) at 37 °C in a humidified atmosphere of 5% CO_2_ with medium changes every 48 to 72 h.

#### 4.2.1. Concentration-Response Assays for rTBL-1 Treatment

From confluent cultures, suspensions of 5 × 10^3^ cells per 500 µL of growth medium were seeded in 16 wells of a 24-well plate (Nunclon Δ Thermo Fisher Scientific™, Sigma-Aldrich^®^ Cat. Num. D7039-1CS, Waltham, MA, USA) and incubated under the described conditions. After 72 h, culture synchronization was carried out by replacing the growth medium with RPMI-1640 supplemented with 2% FBS + 1% Anti-anti for the CT-26 cells and DMEM supplemented with 2% FBS + 1% non-essential amino acid solution + 10 mM Hepes and 1% Anti-Anti for MC-38 cells. Although FBS restriction is a recognized cause of affectation in cell viability [78], its implication was assumed during dose–response assays to ensure the synchronization of the cell cycle prior to the experimental management [79]. After 24 h of incubation in synchronization medium, initial control counts were performed, incorporating 500 µL of trypsin + EDTA solution (GIBCO^®^ Cat. Num. 25300062) to two of the wells and incubating for 15 min (CT-26 cells) or 5 min (MC-38 cells) at 37 °C. Once the enzyme was inactivated with 500 µL of the appropriate growth medium, cell suspensions were transferred to 1.5 mL conical tubes, and 20 µL was charged into each of the two chambers of a Neubauer cell counter (BLAUBRAND^®^ Sigma-Aldrich^®^ Cat. Num. BR717805-1EA, Essex, CT, USA) in order to proceed with the direct count in duplicate. At this point, a second change of conditions was made to the rest of the cultures in the plate by replacing the synchronization medium of each pair of wells with one of the assay mediums consisting of RPMI-1640 or DMEM (with 1% non-essential amino acid solution + 10 mM Hepes), supplemented with 1% anti-anti + 0.5% (*w*/*v*) bovine serum albumin (BSA; GIBCO^®^ Cat. Num. 9048-46-8) and 1 of 6 concentrations of *r*TBL-1 (10, 20, 30, 40, 50, and 100 µg/mL) or vehicle (phosphate-buffered saline, PBS).

After 8 h of exposure, the treated cultures were rinsed with PBS and trypsinized in order to proceed to cell counting as described. The survival percentage was calculated with respect to the mean of the initial control counts. Non-linear regressions were performed for each cell type from the means of 3 independent experiments in duplicate [80], developed with cells from the 4th, 5th, and 6th transfer, to determine the lethal concentration 50 (LC_50_) of each cell type.

#### 4.2.2. Evaluation of Cell Apoptosis by Annexin V Binding Assay

To evaluate the effect of *r*TBL-1 on apoptosis of CT-26 and MC-38 cells, the Annexin V & Dead Cell Kit Muse™ apoptosis detection kit (Cytek^®^ Biosciences Cat. Num. MCH100105, Freemont, CA, USA) was used following the manufacturer’s instructions and using the correspondent *r*TBL-1-LC_50_ g. After treatment, the cultures were rinsed with PBS, followed by trypsinization, and transferred to sterile 15 mL conical bottom tubes (Corning^®^ Cat. Num. 430790) and centrifuged at 125× *g*. Each cell pellet was resuspended in 2 mL of the corresponding assay medium, ensuring complete disintegration of the cell pellet in the suspension. Next, 100 μL of each cell suspension was transferred to a 1.5 mL Eppendorf^®^ tube and 100 μL of Annexin-V reagent was added. The mixture was shaken at medium speed in a vortex (Scientific Industries, INC. G-560, Bohemia, NY, USA) for 3 s and finally, incubated in the dark at room temperature for 15 min. The cell death and survival parameters were quantified in the Muse Cell Analyzer (Millex Millipore^®^ Cat. Num. 0500-3115 Cork, Ireland). Following the manufacturer’s instructions: cells with a positive signal for annexin V and negative for 7-aminoactinomycin D (7-AAD) undergo early apoptosis; annexin V- and 7-AAD-positive cells undergo late apoptosis; a positive stain for 7-AAD and negative for Annexin V was indicative of cell necrosis. For their part, cells that do not present Annexin V or 7-AAD signal are alive and do not undergo quantifiable apoptosis.

#### 4.2.3. Western Blot for Total Caspase-3 and PARP1

Confluent cultures of CT-26 and MC-38 were treated as described (2.2.2) in order to obtain protein extracts and perform Western blot series for caspase-3 and cleaved PARP1 (cPARP1) proteins. Briefly, once the treatment medium was discarded, two rinses with ice-cold PBS were performed. Next, the cell monolayers were scraped (Bio-Rad Laboratories^®^ Gel Release Cat. Num. 1653320, Hercules, CA, USA) and carefully transferred to sterile 15 mL conical bottom tubes in an ice bath in order to subject them to centrifugation (200× *g* for 5 min, at 4 °C). Once the supernatants were discarded, each pellet was resuspended in 200 μL (1 mL per ≈ 1 × 10^8^ cells) of cell lysis buffer (NP40, Invitrogen™ Cat. Num. FNN0021 ÍDEM), previously ice-cold and supplemented with cocktail of protease and phosphatase inhibitors in a 1:9 ratio (Abcam^®^ Cat. Num. ab201119, Cambridge, UK). Subsequently, 70 µg of each protein extract was resolved by SDS-PAGE in a 10% polyacrylamide gel at an initial voltage of 70 V for 15 min and continuing at 120 V until the point at which the running front reached the bottom of the gel. After completing the electrophoresis, the proteins were transferred at a constant value of 200 mAh for 60 min to nitrocellulose membranes as solid support (Thermo Scientific™ Cat. Num. No. Cat. LC2001, Rockford, IL, USA). Once completed, membranes were retrieved with inert forceps and transferred to a shallow dish with tris-glycine transfer buffer (Invitrogen Novex Cat. Num. LC3675, Carlsbad, CA, USA) and they were blocked using a 3% blocking-grade skim milk (Bio-Rad Laboratories^®^ Cat. Num. 1706404EDU) for 1 h. After three TTBS rinses of 10 min each, the membranes were tested by Western blot analysis using the antibodies Anti-caspase-3, clone 3J16 (ZooMAb^®^ Sigma-Aldrich^®^ Cat. Num. ZRB1221, Darmstadt, Germany), and recombinant anti-PARP1 (Abcam^®^ Cat. Num. ab191217, Hangzhou, China) in a concentration of 1:1000 (blocking grade skim milk-3% in TTBS), and as secondary antibody, peroxidase-conjugated Anti rabbit IgG (H+L) (Jackson Immunoresearch Lab. Inc. Cat. Num. 111-035-003, West Grove, PA, USA) in a concentration of 1:10,000 in the same diluent. As a loading control, anti-GAPDH (SANTA CRUZ BIOTECHNOLOGY, INC. Cat. Num. sc-32233, Heidelberg, Germany) was used (1:1000), together with a secondary antibody, peroxidase-conjugated Anti mouse IgG (H+L) in donkey (Jackson Immunoresearch Lab. Inc. Cat. Num. 715-035-150, West Grove, PA, USA) in a concentration of 1:10,000. Finally, the target proteins were detected by SuperSignal West Dura Extended Life Chemiluminescent Substrate (Thermo Scientific™ Cat. Num. 34075, Rockford, IL, USA) and photographed (Thermo Scientific™ CL-XPosure Film Cat. Num. 34090, Rockford, IL, USA) following the manufacturer’s instructions.

### 4.3. Experimental Animals

A total of 14 female BALB/c wild-type (+/+) mice and 14 females BALB/c nude (*nu*/*nu*) mice homozygous for MHC haplotype H2^d^ (Charles River Laboratories International, Inc., Wilmington, MA, USA), all 4- to 6 weeks old, were purchased from Circulo ADN S.A. of C.V., (Iztapalapa, Mexico City, Mexico). Upon receipt, the immunocompetent mice were housed in the vivarium of the Faculty of Natural Sciences of the Autonomous University of Queretaro and cared for according to the Official Mexican Standard (NOM) 062-ZOO-1999 under the following parameters: inverted 12:12 h light/dark cycles, ambient temperature of 18 to 26 °C, relative humidity of 40–70%, and changes of sanitary bedding when appropriate. The immunodeficient mice were housed in a specific pathogen-free (SPF) environment in the research tower of the National Cancer Institute of Mexico under 12:12 h light/dark cycles in autoclaved cages and provided with sterilized bedding under controlled conditions of temperature (23 °C) and humidity (40–70%), as well as >15 air changes per h. All animals underwent a two-week acclimatization period before the start of the experiments and were fed ad libitum with purified water and a high-protein autoclaved food at all times. The Bioethics Committee of the Faculty of Chemistry of the Autonomous University of Querétaro approved all experimental procedures (CBQ21/065).

#### 4.3.1. Establishment of CT-26 Colon Cancer Cell Tumor Allografts

A total of 5 × 10^4^ CT-26 cells (in 50 µL of PBS) were implanted subcutaneously (s.c.) into the shaved-lateral flank panicle of the wild-type BALB/c mice, using a 25G × 1 ½ in needle. Once the tumor allografts reached volumes between 25 and 50 mm^3^, animals were randomly assigned to either the vehicle-treated control group (PBS) or the *r*TBL-1 treatment group (6 mg per kg of body weight dissolved in PBS), ensuring the matching of the mean initial tumor volume between the experimental groups. Those individuals who developed more than one tumor were discarded. The development of the ectopic tumor allografts was monitored by direct measurement (digital vernier caliper, Gyros DIGI-Science Accumatic SKU EO-254, Monsey, NY, USA) every 72 h. To avoid animal suffering, those wild-type BALB/c specimens that reached volumes greater than 1000 mm^3^ and/or showed signs of necrosis in the tumor mass were euthanized by asphyxiation in a carbon dioxide chamber. For BALB/c (*nu*/*nu*) mice, the volume for the endpoint was determined at ≥2000 mm^3^ due to the greater resistance of their looser, thicker, and hyperplastic skin, typical of the phenotype FOXN1 with loss-of-function mutations [81]. The volume of each tumor and the tumor growth inhibition (TGI) were, respectively calculated using the following formulas:0.5 × length × width^2^
where length was the longest dimension.[(C_t_ − C_0_) − (T_t_ − T_0_)] / (C_t_ − C_0_) × 100
where C_t_ was the mean tumor volume in the control group at time (*t*); C_0_ was the mean tumor volume in the control group at *t*_0_; T_t_ was the mean tumor volume in the treatment group at *t*; and T_0_ was the mean tumor volume in the treatment group at *t*_0_ [82].

#### 4.3.2. Evaluation of *r*TBL-1 Antitumor Activity In Vivo

To examine the antitumor activity of *r*TBL-1, we studied the effect of its intralesional administration on the growth of preexisting palpable tumors in BALB/c mice. Briefly, 24 h after randomization of individuals, 50 µL of PBS (vehicle) was injected into the tumor of 7 BALB/c individuals, and in parallel, one dose of *r*TBL-1 (in the same volume of vehicle) was injected into the tumor of an equal number of individuals, in a schedule of four doses in total, every five days. The diameter of each tumor was measured longitudinally every 72 h.

#### 4.3.3. Evaluation of T Cell Dependence in the Antitumor Activity of *r*TBL-1

In order to determine whether the therapeutic activity of *r*TBL-1 was dependent on host T cells, 14 BALB/c (*nu*/*nu*) mice were subjected to the tumor inoculation scheme previously described, as well as to treatment with *r*TBL-1 (n = 7) in the same administration schedule. The treatment effect was evaluated with respect to a control group administered with vehicle (n = 7).

### 4.4. Statistical Analysis

Dose–response curves were analyzed by non-linear regression and, as with the experimental flow cytometry data and Western blot densitometry, were compared by one-way ANOVA using the means of three or four independent experiments by Dunnett’s or Tukey’s post hoc tests using GraphPad Prism 8.0.1 (grouping of information by Tukey’s method was performed in Minitab^®^ 18.1). Densitometric quantification of Western blots was performed using ImageJ 1.54g software [83]. The comparison of in vitro treatment responses between both cell types was performed by two-sample Student’s *t*-test using Minitab^®^ 18.1. From in vivo assays, tumor allograft development was compared by ratio paired *t*-test, and survival data were analyzed by Kaplan–Meier curve and log-rank (Mantel–Cox) test using GraphPad Prism 8.0.1. For all tests, a *p* value < 0.05 was considered statistically significant.

## 5. Conclusions

Our results indicate that the ability of *r*TBL-1 to induce apoptosis in colon cancer cells does not only depend on the availability of EGFR in the tumor cell, given the existence of unexplored mechanisms independent of this receptor. The expression of EGFR could be associated with deeper responses, possibly related to a higher level of *r*TBL-1 binding to the cell surface and the consequent cross-linking of glycosylated receptors yet to be identified. The benefit of *r*TBL-1 treatment was dependent on host’s T cells, which might indicate that the antitumor effect of *r*TBL-1 constitutes the synergistic result of apoptosis induction and T cell-mediated cytotoxicity in the tumor. Future studies will focus on elucidating the immunogenic effects triggered by the antitumor activity of *r*TBL-1 in colon cancer.

## Figures and Tables

**Figure 1 pharmaceuticals-18-00213-f001:**
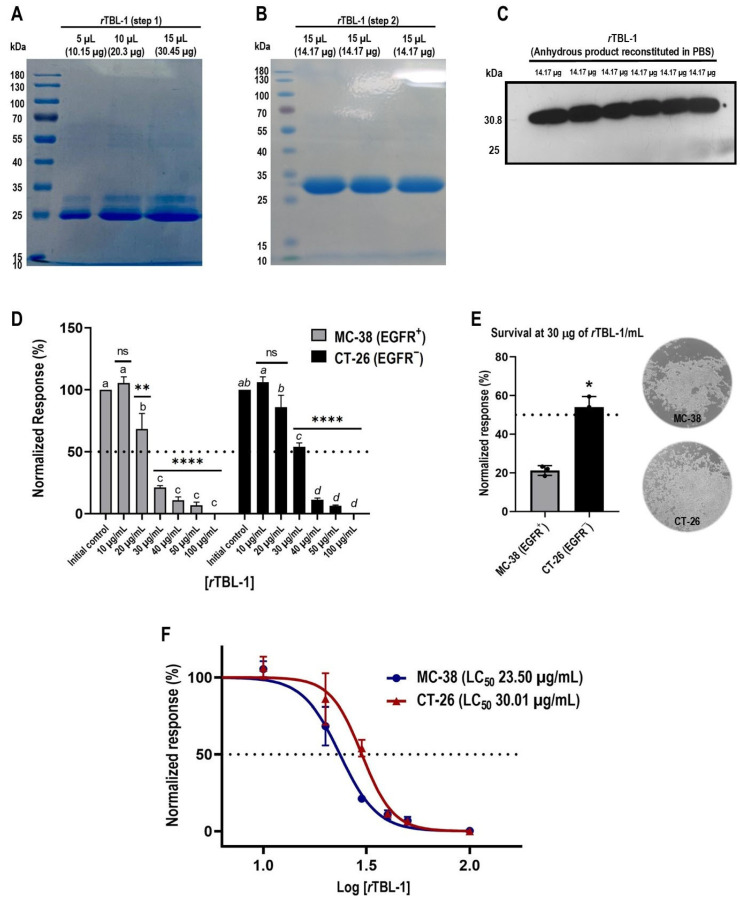
*r*TBL-1 produces cytotoxic effects in colon cancer cells in a concentration-dependent manner. (**A**) A 10% SDS-PAGE analysis confirms the presence of a protein band with a molecular weight of ~30.8 kDa, together with another that constitutes a major impurity (25 kDa). (**B**) A second purification step with nickel affinity chromatography allows the removal of the contaminating protein in correlation with the enrichment of the target band. (**C**) The identity of *r*TBL-1 is confirmed by Western blot for 6xHis-tag. (**D**) Concentration–response assays reveal response patterns at 8 h of *r*TBL-1 exposure in both cell types (Dunnett’s and Tukey’s post hoc tests). (**E**) Treatment tolerance between cell lineages at 30 µg *r*TBL-1/mL (Student’s *t-*test) and representative full-field micrographs (10×) show the post-treatment status of the cultures. (**F**) Dose–response curves by non-linear regression demonstrate that MC-38 EGFR^+^ and CT-26 EGFR^−^ colon cancer cells were responsive to treatment with *r*TBL-1, presenting LC_50_ values of 23.50 and 30.01 µg of *r*TBL-1/mL, respectively (*p* = 0.063). Data are presented as mean ± standard deviation from three independent experiments performed in duplicate. One-way ANOVA with post hoc tests; for Dunnett, * *p* < 0.05, ** *p* < 0.005, and **** *p* < 0.0001; and for Tukey, those means that do not share a letter are significantly different; ns indicates non-significant statistical differences.

**Figure 2 pharmaceuticals-18-00213-f002:**
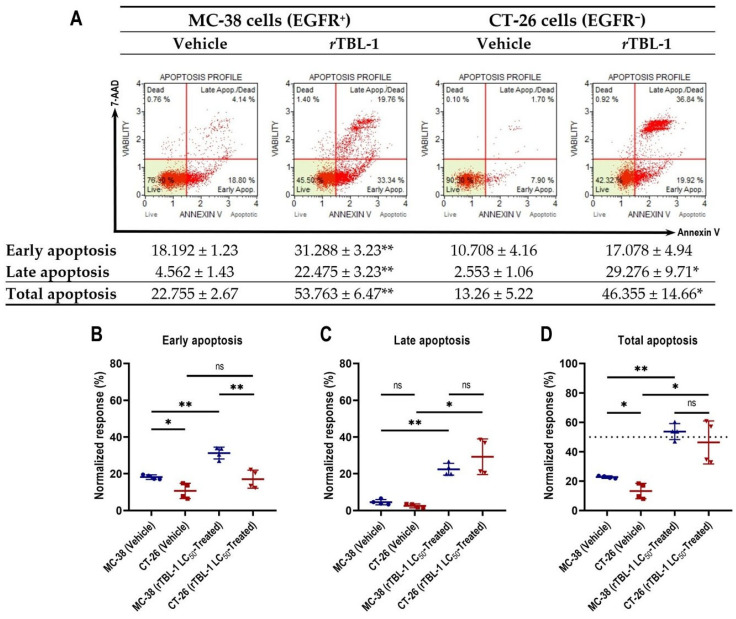
*r*TBL-1-induced apoptosis is slower and longer-lasting in MC-38 EGFR^+^ cells than in CT-26 EGFR^−^ cells. (**A**) Representative dot plots of apoptosis analysis by flow cytometry for MC-38 and CT-26 cells exposed for 8 h to vehicle or its corresponding *r*TBL-1-LC_50_, supplemented with 0.5% BSA (*w*/*v*) instead of FBS. Detailed analysis of apoptotic kinetics for MC-38 and CT-26 cells in (**B**) early, (**C**) late, and (**D**) total apoptosis. Data are presented as mean ± standard deviation from four independent experiments. Student’s two-sample *t*-test, * *p* < 0.05, ** *p* < 0.005 indicate significant differences with respect to control cells; ns indicates non-significant statistical differences.

**Figure 3 pharmaceuticals-18-00213-f003:**
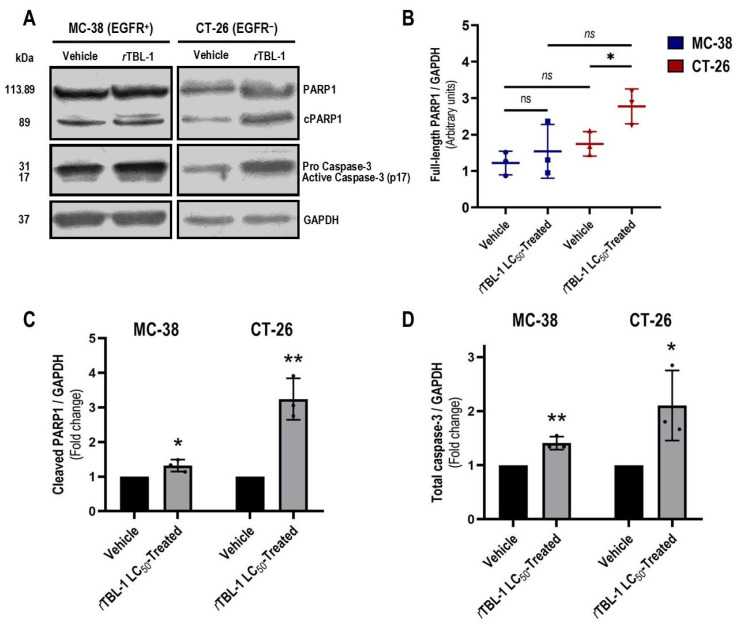
Exposure to *r*TBL-1 produces increases in caspase-3 proteolytic activation and PARP1 cleavage in both EGFR^+^ and EGFR^−^ murine colon cancer cells. (**A**) The effects of *r*TBL-1 treatment on apoptosis markers were analyzed by Western blot after 8 h of exposure at the corresponding LC_50_, and GAPDH was used as a loading control. (**B**) CT-26 cells increased their levels of full-length PARP1 in a rheostatic fashion to treatment (Student’s *t-*test). (**C**) Treatment-induced PARP1 cleavage levels in MC-38 and CT-26 cells. (**D**) Treatment-induced levels of procaspase-3 and caspase-3 in MC-38 and CT-26 cells. Quantification data are presented as mean ± standard deviation of the densitometry data of Western blots with indicated antibodies on extracts from three independent experiments. One-way ANOVA with Dunnett’s post hoc test, * *p* < 0.05, ** *p* < 0.005; ns indicates non-significant statistical differences.

**Figure 4 pharmaceuticals-18-00213-f004:**
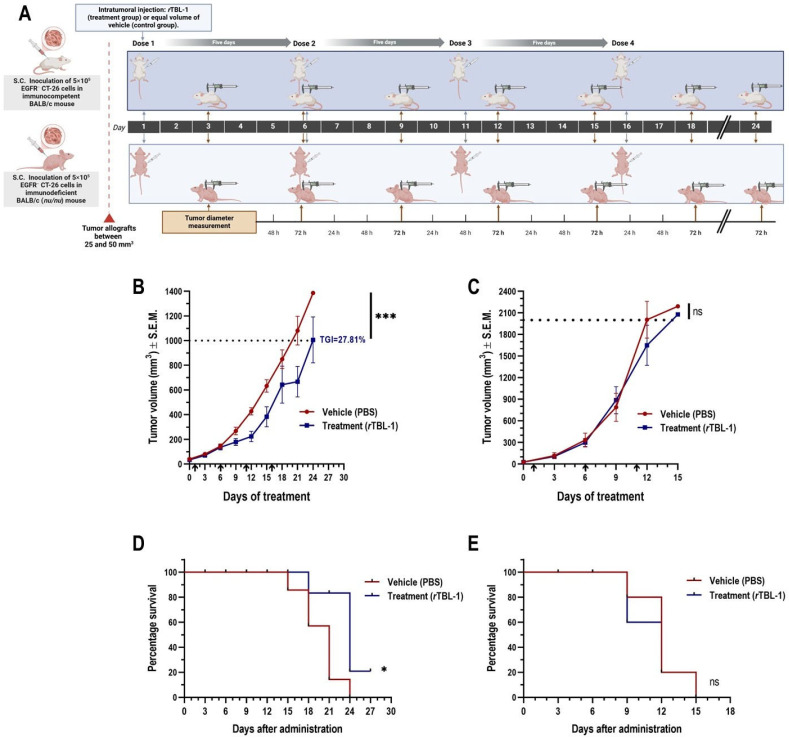
Intralesional administration of *r*TBL-1 inhibits tumor growth of EGFR^−^ syngeneic colon cancer CT-26 in immunocompetent BALB/c (+/+) mice but not in immunodeficient BALB/c (*nu*/*nu*) mice. (**A**) Experimental design for proof of concept of intralesional *r*TBL-1 in the treatment of CRC tumors and the possible involvement of the host immune system. Effect of treatment on tumor volume in (**B**) immunocompetent BALB/c (+/+) and (**C**) immunodeficient BALB/c (*nu*/*nu*) hosts. Tumor volume is represented as mean ± S.E.M. of five to seven animals per group. The percentage of TGI is presented when the difference in mean tumor volume of the treatment group reached statistical significance compared to that of the control group. Dotted lines indicate the endpoints defined for each mouse strain. Arrows indicate time points of each administration. Ratio paired *t*-test, * *p* < 0.05 and *** *p* < 0.0005. Analysis of the effect of treatment on endpoint survival using Kaplan–Meier plots in (**D**) immunocompetent BALB/c (+/+) and (**E**) immunodeficient BALB/c (*nu*/*nu*) hosts. Log-rank (Mantel–Cox) test, * *p* < 0.05; ns indicates non-significative statistical differences. Figure of experimental design was elaborated using the BioRender platform (https://biorender.com).

**Table 1 pharmaceuticals-18-00213-t001:** Mutational statuses of the main molecular markers that govern therapeutic choice in mCRC in *r*TBL-1-responsive colon cancer cell lines.

	Mutational Status		
Cancer Cell Line (Organism/Tissue of Origin)	EGFR	KRAS	BRAF
HT-29 (Human colorectal adenocarcinoma)	Wild type [45] Normally expressed [46]	Wild type [51]	Heterozygous mutation^V600E^ [52,53]
2. SW-480 (Human colorectal adenocarcinoma Dukes’ type B)	Wild type, overexpressed [42]	Mutated^G12V^ [54,55,56]	Wild type [52]
3. CT-26 (Murine colon carcinoma)	Not expressed [30]	Mutated^G12D^ [30,56]	Wild type [30]
4. MC-38 (Murine colon adenocarcinoma)	^1^ Mutated^G1103C^ [29,47] Overexpressed [57]	Wild type [39]	Mutated^W487C^ [39,58]

^1^ Passenger mutation.

## Data Availability

Data are contained within the article.

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
