# Peer review of "Phaseolus acutifolius Recombinant Lectin Exerts Differential Proapoptotic Activity on EGFR+ and EGFR− Colon Cancer Cells and Provokes T Cell-Assisted Antitumor Responses in Mice"

_pharmaceuticals, 2025, doi:10.3390/ph18020213_

Round 1
Reviewer 1 Report
Comments and Suggestions for Authors
After reading and analyzing the manuscript ,, Phaseolus acutifolius recombinant lectin exerts differential 2 proapoptotic activity on EGFR+ and EGFR− colon cancer cells 3 and provokes T cell-assisted antitumor responses in mice”, I find that:
· The study investigates the therapeutic potential of rTBL-1, a recombinant lectin from Phaseolus acutifolius, focusing on its impact on colon cancer cells and tumorigenesis in vivo;
· The research reveals intriguing insights into its mechanism of action and possible immunogenic properties.
Here are some specific points for improvement:
- Figure 1: Figures should provide more descriptive explanations. For example, in Figure 1D, clarification is needed regarding the meaning of "abcd." Please specify what each part represents or signifies.
- Figure 2A: Please justify the choice of One-way ANOVA with Dunnett's post hoc test for the analysis.
- Lines 281-283: Please provide detailed information on how modulation of the EGFR pathway correlates with apoptosis of SW-480 cells treated with rTBL-1. Elaborate on the association with phosphorylation (Tyr1068) of the receptor, involving phospho-EGFR / phospho (Thr180/Tyr182)-p38 / phospho (Ser15)-p53.
- Additional Information: Include more details on the oncogenic alteration of EGFR downstream effectors, emphasizing their deregulation, which leads to uncontrolled signaling and cellular proliferation independent of the receptor and its ligand.
- References: Please ensure that all references are accompanied by the appropriate page numbers.
Author Response
Comments and Suggestions for Authors
After reading and analyzing the manuscript, Phaseolus acutifolius recombinant lectin exerts differential 2 proapoptotic activity on EGFR+ and EGFR− colon cancer cells 3 and provokes T cell-assisted antitumor responses in mice”, I find that:
- The study investigates the therapeutic potential of rTBL-1, a recombinant lectin from Phaseolus acutifolius, focusing on its impact on colon cancer cells and tumorigenesis in vivo;
- The research reveals intriguing insights into its mechanism of action and possible immunogenic properties.
Here are some specific points for improvement:
- Figure 1: Figures should provide more descriptive explanations. For example, in Figure 1D, clarification is needed regarding the meaning of "abcd." Please specify what each part represents or signifies.
Small letters “abcd” represents significative statistical difference We have incorporated this note in the caption of Figure 1D (line 154) and for clarity, we have added some key descriptions in the whole Figure legend (lines 148-161).
- Figure 2A: Please justify the choice of One-way ANOVA with Dunnett's post hoc test for the analysis.
Thank you for your comment. At first, we determined the statistical significance between three groups for each cell line: the control group (cultures with cell cycle synchronization, treated with vehicle), a treatment group (also synchronized and treated with rTBL-1) and, as part of the validation of the method, a third group corresponding to cells in growth conditions (without prior synchronization of the cell cycle). Therefore, the Dunnett post hoc test was chosen to complete the analysis. However, the results of the third group were not considered relevant to the article. As you have made us see, the final results present only the comparison between two groups (control vs treated), hence we modified the analysis and used a two-sample t-test instead a one-way ANOVA. Modifications were done in lines 124, 127, 128 and 173.
- Lines 281-283: Please provide detailed information on how modulation of the EGFR pathway correlates with apoptosis of SW-480 cells treated with rTBL-1. Elaborate on the association with phosphorylation (Tyr1068) of the receptor, involving phospho-EGFR / phospho (Thr180/Tyr182)-p38 / phospho (Ser15)-p53.
In lines 284-290: we have developed a detailed description of the work (Dena-Beltrán et al., 2023), relative to the postulated mechanistic axis.
- Additional Information: Include more details on the oncogenic alteration of EGFR downstream effectors, emphasizing their deregulation, which leads to uncontrolled signaling and cellular proliferation independent of the receptor and its ligand.
Here we briefly review the mechanisms underlying the oncogenicity of RAF/BRAF-harbored mutations in cell lines with confirmed responsiveness to rTBL-1:
- BRAFV600E: Mimics the activated conformational state in wild-type BRAF (Cantwell-Dorris et al., 2011).
- KRASG12V / KRASG12D: cause the loss of intrinsic GTPase activity of KRAS, which keeps the protein in a constitutively active state bound to GTP (Vatansever et al., 2020).
- BRAFW487C: Located in the kinase domain of the protein, this is a rare but oncogenically relevant mutation. It is thought to produce a constitutively activated conformational state, similar to that caused by the V600E mutation (Dankner et al., 2018).
Since all the described mutations produce constitutively activated conformational states in their respective proteins, we have made precisions in lines 319-323 and added the previously cited references, reserving the detailed description of the oncogenic mechanisms for the specialized literature.
REFERENCES.
Cantwell-Dorris, E. R., O’Leary, J. J., & Sheils, O. M. (2011). BRAFV600E: Implications for Carcinogenesis and Molecular Therapy. Molecular Cancer Therapeutics, 10(3), 385–394. https://doi.org/10.1158/1535-7163.MCT-10-0799
Dankner, M., Rose, A. A. N., Rajkumar, S., Siegel, P. M., & Watson, I. R. (2018). Classifying BRAF alterations in cancer: new rational therapeutic strategies for actionable mutations. Oncogene 2018 37:24, 37(24), 3183–3199. https://doi.org/10.1038/s41388-018-0171-x
Vatansever, S., Erman, B., & GümüÅŸ, Z. H. (2020). Comparative effects of oncogenic mutations G12C, G12V, G13D, and Q61H on local conformations and dynamics of K-Ras. Computational and Structural Biotechnology Journal, 18, 1000. https://doi.org/10.1016/J.CSBJ.2020.04.003
- References: Please ensure that all references are accompanied by the appropriate page numbers.
All references have been completed.
Thank you very much, your observations were very important in order to improve the manuscript.
Reviewer 2 Report
Comments and Suggestions for Authors
I am pleased to review the manuscript titles as “Phaseolus acutifolius recombinant lectin exerts differential proapoptotic activity on EGFR+ and EGFR− colon cancer cells and provokes T cell-assisted antitumor responses in mice”
This research articles tried to elucidated the dual mechanism of action of rTBL-1, a recombinant lectin with promising antitumor activity, in colon cancer. The findings demonstrate that rTBL-1 induces apoptosis in colon cancer cells independently of EGFR. Furthermore, the study highlights the role of adaptive immunity in rTBL-1- antitumor efficacy, as evidenced by its selective activity in immunocompetent mice (27.81% tumor inhibition) and lack of response in T cell-deficient models. It targets both EGFR+ and EGFR− tumors making it a versatile candidate for translational development in heterogeneous colon cancers.
I do believe the topic is very interesting and experiments are reported in very detailed and reproducible manners. However, I have some suggestions that in my opinion may prove beneficial for the further improvement of this manuscript.
# In abstract (lines 22-25) consider splitting complex sentences for improved flow and comprehension
# While LC50 values for EGFR+ (23.50 µg/mL) and EGFR− (30.01 µg/mL) cells are presented, the abstract does not address whether this difference is statistically significant.
# A 27.81% tumor growth inhibition in immunocompetent mice is reported, but statistical significance (e.g., p-values) and survival benefit metrics (e.g., hazard ratios) are absent. Include these to strengthen claims of efficacy.
# I would like to see the acknowledgement of limitations of the study in discussion section.
Best wishes...
Author Response
Comments and Suggestions for Authors
I am pleased to review the manuscript titles as “Phaseolus acutifolius recombinant lectin exerts differential proapoptotic activity on EGFR+ and EGFR− colon cancer cells and provokes T cell-assisted antitumor responses in mice”
This research articles tried to elucidated the dual mechanism of action of rTBL-1, a recombinant lectin with promising antitumor activity, in colon cancer. The findings demonstrate that rTBL-1 induces apoptosis in colon cancer cells independently of EGFR. Furthermore, the study highlights the role of adaptive immunity in rTBL-1- antitumor efficacy, as evidenced by its selective activity in immunocompetent mice (27.81% tumor inhibition) and lack of response in T cell-deficient models. It targets both EGFR+ and EGFR− tumors making it a versatile candidate for translational development in heterogeneous colon cancers.
I do believe the topic is very interesting and experiments are reported in very detailed and reproducible manners. However, I have some suggestions that in my opinion may prove beneficial for the further improvement of this manuscript.
# In abstract (lines 22-25) consider splitting complex sentences for improved flow and comprehension
The proposed lines have been corrected for better understanding.
# While LC50 values for EGFR+ (23.50 µg/mL) and EGFR− (30.01 µg/mL) cells are presented, the abstract does not address whether this difference is statistically significant.
The difference in LC50 between cell lines was not statistically significant (p = 0.063). We have incorporated the p value in the summary (line 24), as well as in lines 158 and 300.
# A 27.81% tumor growth inhibition in immunocompetent mice is reported, but statistical significance (e.g., p-values) and survival benefit metrics (e.g., hazard ratios) are absent. Include these to strengthen claims of efficacy.
The p-value for the tumor growth inhibition assay is presented on lines 249 and 403. We also have incorporated the p-value into the abstract (line 28) and added the 95% confidence interval for the ratio (line 249). Similarly, the p-value for the survival analysis is presented on lines 252 and 403. After some adjustments to the abstract, we also incorporated it on line 29. We also added the hazard ratio (logrank method) on line 252.
# I would like to see the acknowledgement of limitations of the study in discussion section.
In lines 418 to 425 we discuss the limitations faced during the execution of this work, which relate to the impossibility of developing analyses that would confirm the postulated anti-tumor immune response, as well as technical difficulties of the model and its impact on the number of individuals per experimental group.
Thank you very much, your observations were very important in order to improve the manuscript.